# Effect of Ivabradine on Cardiac Ventricular Arrhythmias: Friend or Foe?

**DOI:** 10.3390/jcm10204732

**Published:** 2021-10-15

**Authors:** Marta Oknińska, Aleksandra Paterek, Zuzanna Zambrowska, Urszula Mackiewicz, Michał Mączewski

**Affiliations:** Centre of Postgraduate Medical Education, Department of Clinical Physiology, ul. Marymoncka 99/103, 01-813 Warsaw, Poland; marta.okninska@cmkp.edu.pl (M.O.); aleksandra.paterek@cmkp.edu.pl (A.P.); zuzanna.zambrowska@cmkp.edu.pl (Z.Z.); urszula.mackiewicz@cmkp.edu.pl (U.M.)

**Keywords:** ivabradine, HCN channels, heart failure, coronary artery disease, ventricular arrhythmias, ventricular tachycardia, cardiac myocytes

## Abstract

Life-threatening ventricular arrhythmias, such as ventricular tachycardia and ventricular fibrillation remain an ongoing clinical problem and their prevention and treatment require optimization. Conventional antiarrhythmic drugs are associated with significant proarrhythmic effects that often outweigh their benefits. Another option, the implantable cardioverter defibrillator, though clearly the primary therapy for patients at high risk of ventricular arrhythmias, is costly, invasive, and requires regular monitoring. Thus there is a clear need for new antiarrhythmic treatment strategies. Ivabradine, a heartrate-reducing agent, an inhibitor of HCN channels, may be one of such options. In this review we discuss emerging data from experimental studies that indicate new mechanism of action of this drug and further areas of investigation and potential use of ivabradine as an antiarrhythmic agent. However, clinical evidence is limited, and the jury is still out on effects of ivabradine on cardiac ventricular arrhythmias in the clinical setting.

## 1. Ivabradine—Brief Summary

Ivabradine is a presumably selective heartrate-reducing agent, the only inhibitor of HCN channels available for human use, currently approved for the symptomatic treatment of chronic coronary artery disease (CAD) and chronic heart failure (CHF) in patients with heart rate ≥ 70 bpm in combination with standard therapy [1,2,3].

## 2. HCN Channel Family—A Primary Target for Ivabradine

Hyperpolarization-activated, cyclic-nucleotide-gated channels form a family of nonselective cation channels conducting mainly sodium and potassium ions through the plasma membrane and generating a current termed *I*_f_ (for “funny“). They have a unique biophysical behavior, i.e., they open upon hyperpolarization and result in an inward, depolarizing, predominantly sodium current and hence the name of the current, “funny” [4]; moreover, they are deactivated by depolarization. Additionally they are activated by cyclic nucleotides, cAMP and cGMP [5] and thus they are called “hyperpolarization-activated, cyclic-nucleotide-gated” (HCN) channels.

Functional HCN channels result from the assembly of four either identical or different alpha subunits. There are four different genes coding for HCN subunits (1–4) [6]. All HCN monomers (HCN1, HCN2, HCN3 and HCN4) have the same fundamental structure: contain a transmembrane core with six alpha helices (S1–S6): a voltage-sensing domain (S1–S4) and a pore region (S5–S6) carrying the ion selectivity filter as well as two cytosolic domains at the NH and COOH termini [7]. The elements conferring ion selectivity and voltage sensitivity are located in the transmembrane core and show high degree of sequence homology (80–90%) within the HCN family [8], while the NH and COOH termini vary in length and amino acidic composition. The COOH terminus harbors the cyclic-nucleotide-binding domain (CNBD), which is critical for the modulation by cyclic nucleotides [9].

HCN channels are voltage-gated channels activated at negative potentials. Their half activation voltage (V_1/2_) ranges from −70 mV for HCN1 to −100 mV for HCN4, while HCN2 and HCN3 fall in the middle of this range [10]. Gating kinetics are also voltage dependent and the activation time constant is inversely related to the hyperpolarization. The activation time constant is the fastest in HCN1 (25 ms), the slowest in HCN4 (200–500 ms), while HCN2 and HCN3 again lie in between [11]. Relative permeability of HCN to Na^+^ and K^+^ is 1:3/1:5 and the reversal potential is −25 to −40 mV, but due to the negative V_1/2_, HCN channels conduct primarily an inward Na^+^ current [10]. Binding of cAMP to the CNBD shifts upward the activation curve and accelerates the activation time constant, but is not sufficient to open the channels without an additional hyperpolarization of the membrane. Sensitivity to cAMP is highest in HCN2 and 4, weak in HCN1, while HCN3 exhibits no cAMP sensitivity [10].

## 3. HCN Channels in the Heart—Expression and Role

HCN channels are essential for normal impulse generation and conduction in the heart. In the sinoatrial node (SAN) cells they form a specialized pacemaking system involving HCN cooperation with ryanodine receptors (RyRs) of sarcoplasmic reticulum (SR) that is responsible for SAN spontaneous electrical activity [12]. HCN channels together with T- and L-type calcium channels as well as sodium-calcium exchanger (NCX) form the membrane clock, while RyRs in SR form the calcium clock. These two clocks constitute a tightly connected pacemaking system that is eventually responsible for spontaneous firing of SAN cells [13].

At maximally negative diastolic potentials (from −70 to −40 mV depending on SAN region), large number of HCN channels are open and an inward current flowing through them slowly depolarizes the membrane of SAN cells during the diastolic phase (Figure 1) [12]. Concomitantly submembrane Ca^2+^ wavelets appear as Ca^2+^ is released by RyRs from SR. This leads to NCX activation and further depolarizing membrane current. Eventually the threshold required to activate membrane L-type calcium channels is reached and a spontaneous action potential is generated. Finally Ca^2+^ inflow through L-type calcium channels triggers final Ca^2+^ release from SR [13].

In the atrioventricular node HCN channels are involved in generation of spontaneous electrical activity and conduction of stimuli from atria to ventricles [14]. cAMP-dependent regulation of HCN channel activity plays a key role in the control of SAN firing rate and consequently heart rate as well as AV node conduction by the autonomic nervous system.

HCN channels, primary site of action of ivabradine, are highly expressed in the sinoatrial node, while their expression in normal ventricular myocytes is low (left panel). In the sinoatrial node *I*_f_ current flowing through HCN channels is responsible for slow diastolic depolarization (indicated by a horizontal blue bar) toward the threshold required to activate calcium channels and generate a spontaneous action, thus participating in the pacemaking activity of sinoatrial node (black curve). Ivabradine inhibits *I*_f_ current and slows the diastolic depolarization, leading to heart rate reduction (red curve). Antiarrhythmic effects of ivabradine in the setting of ischemia are probably related to heart rate reduction. HCN channels are weakly expressed in normal ventricular myocytes. Limited evidence indicates that low doses of ivabradine, acting primarily on HCN channels, results in mild hyperpolarization and shortening of action potential duration (APD). Physiological relevance of this effect is unknown, but may include mild increase of spatial dispersion of repolarization. High doses of ivabradine can also inhibit I*_Kr_* current, resulting in prolongation of APD; this could be antiarrhythmic in the setting of short QT (e.g., induced by digitalis), but proarrhythmic in the setting of long QT (e.g., induced by class III antiarrhythmic drugs). Expression of HCN channels in the ventricular myocytes is increased in heart failure. This results in diastolic cardiomyocyte depolarization and reduced amplitude of action potential overshoot. In such setting *I*_f_ current may be responsible for abnormal automaticity (caused by an inward current generated by overexpressed HCN channels) in the ventricular cardiomyocytes, resulting in ventricular premature complexes that may trigger ventricular arrhythmias. Ivabradine may prevent it by abolishing HCN channel overexpression and HCN channel blockade. 

In the heart all the four isoforms have been detected and their expression varies according to the cardiac region, although HCN4 accounts for approximately 80% of total HCN proteins. In the adult heart the expression of HCN channels is largest in the SAN region and in the conduction system (atrioventricular node and Purkinje fibers), while their expression in the atria and ventricles is low [15]. HCN4 is the main isoform in the SAN (being responsible for 70–80% of total *I*_f_ current), atrioventricular node and Purkinje fibers [14], while HCN2 and HCN4 are the most abundant isoforms in the mammalian atria and ventricles and together represent more than 70–90% of the ventricular HCN channels [16], HCN3 accounting for the rest. HCN4 is also thought to play a crucial role in autonomic control of heart rate [17]. On the other hand HCN1 is almost exclusively present in SAN, being quite specific molecular marker of the latter [18].

At early embryonic stages HCN4 is abundantly expressed in the whole heart [19] and is largely responsible for *I*_f_ triggered automaticity of ventricular myocytes [20] as well as being involved in ventricular wall maturation at embryonic stages [21]. Toward birth, HCN4 transcription is downregulated in working-type cardiomyocytes and remains at low levels during adult stages [20].

In CHF, cardiac hypertrophy as well as in myocardial infarction (MI), expression of HCN channels in the ventricular cardiomyocytes increases. It was found in explanted human failing hearts [22,23,24], rats after MI [25,26,27], rats and mice with cardiac hypertrophy [16,28,29] and in mouse failing hearts [30,31]. Moreover, these channels are functional, since *I*_f_ current is also increased by twofold to threefold in both rat [22] and mouse [30] ventricular myocytes. The upregulated HCN channels exhibit their normal properties, i.e., they are activated by hyperpolarization and by β-receptor agonists [24]. Mechanism of this effect is unknown, but aldosterone has been shown to promote HCN2 and HCN4 mRNA and protein expression in cultured neonatal rat ventricular myocytes and that aldosterone blocking agents (spironolactone and eplerenone) counteract this effect by upregulating miRNA-1 expression, thus partially contributing to the posttranscriptional repression of HCN4 [32].

On the other hand, streptozocin-induced hyperglycemia resulted in downregulation of ventricular HCN2, while that in the atria was unchanged [33].

There are suggestions that HCN channels in the atria and ventricles prolong AP duration, especially in the epicardial myocytes, resulting additionally in reduction of transmural dispersion of AP duration [34]. APs recorded from epicardial myocytes of HCN3‒/‒ mice were found to be markedly shorter than those of wild-type mice because the depolarizing HCN3-mediated current component was missing. The difference was mainly due to a shortening of the late repolarization phase [34]. HCN knockout prevented hypertrophy-induced QT prolongation [16].

## 4. Effect of Ivabradine on Ion Channels

Ivabradine is the only HCN channel inhibitor approved for human use [35]. It blocks the HCN channel only from the intracellular side when the channels are opened by hyperpolarization [36], thus ivabradine needs an open channel to access its binding site [37] with enhanced binding upon frequent changes in the direction of ion flow [38]. When the channel is closed, the drug molecule is “trapped” in the binding site, which stabilizes the channel blockade [37]. Ivabradine blocks all four HCN isoforms with equal potency [39].

Ivabradine is a relatively selective inhibitor of HCN channels compared to other ion channels. Delpon et al. found that the hKv1.5 was blocked by the drug in a concentration-dependent way with IC_50_ 29.0 ± 1.9 μM, therefore at doses higher than those required to block HCN channels [40]. Moreover, in the clinically relevant doses ivabradine did not affect calcium channels [41] and inhibited sodium channels at high concentrations (IC_50_ 30 ± 3 μM) [6,41].

However, recent data indicates that ivabradine also inhibits the rapidly activating delayed rectifier potassium current (*I*_Kr_) (IC_50_ of ≈2–6.8 μM) [41,42,43]. This current is critical for cardiac action potential repolarization and is encoded by the human ether-a-go-go-related gene (hERG). In this context, ivabradine increased action potential duration in human papillary muscles by 11% [44], presumably due to *I*_Kr_ blockade, although at high concentrations (10 µM, while normal ivabradine plasma concentrations in humans following oral dosing 5–20 mg are 0.03–0.13 µM [45]. Similarly it increased action potential duration in an isolated rabbit heart by 16% (at 5 µM), while lower concentrations were ineffective [46] and increased APD in fetal mouse hearts [43]. The fact that in all these studies resting membrane potential was unchanged despite prolongation of APD further corroborates that this effect was mediated by *I*_Kr_ inhibition [41,42,43]. Studies indicate that ivabradine at 10 μM blocks approximately 20–60% of *I*_Kr_ [47] and 80% of HCN4 current [43]. However, the European Medicines Agency (EMA) document library (http://www.ema.europa.eu/docs/en_GB/document_library/EPAR_scientific_.Discussion/human/000598/WC500035338.pdf; last accessed on 12 October 2021) indicates that ivabradine blocks the hERG current with an IC50 of 4.85 μM, which is also markedly higher than the reported plasma concentrations in humans. So there seems to be a consistent discrepancy between in vivo and in vitro responsiveness to ivabradine probably caused by high lipophilicity of ivabradine and its preferential accumulation in the plasma membranes, ensuring easy access to all ion channels [43].

hERG mutations or blockage by drugs resulting in reduction of *I*_Kr_ are principal causes of long QT syndrome predisposing individuals to ventricular arrhythmias and sudden cardiac death [48]. This raises concerns over possible proarrhythmic effects of ivabradine through QT prolongation and increased propensity to ventricular tachycardia (of torsades de pointes type), in particular in cases when cardiac repolarization reserve is reduced. This might be the case in early MI, since hypoxia was shown to downregulate hERG through upregulation of calpain mediated cleavage [49] and we indeed demonstrated that hERG abundance is reduced in post-MI rat hearts as early as 24 h after MI induction [25]. 

Furthermore this data suggests that coadministration of ivabradine with drugs known to prolong the QT interval should be approached with caution. Ivabradine-induced bradycardia could potentially exacerbate effects of other drugs on repolarization and there may be synergistic inhibitory effects on hERG. Indeed, there are two recent case reports documenting occurrence of torsade de pointes ventricular tachycardia in the setting of coadministration of ivabradine and azithromycin [50] or ranolazine and diltiazem [51].

## 5. Pathophysiology of Ventricular Arrhythmias

Pathophysiology of vast majority of ventricular arrhythmias involves (1) a trigger and (2) a substrate that propagates the trigger. The importance of a third component, the driver, is probably limited to atrial fibrillation. In specific arrhythmias these slightly vary, but the general pattern is the same.

Arrhythmic triggers involve premature complexes that are induced by triggered activity or automaticity. For example, in selected cases and in specific clinical settings, monomorphic premature ventricular complexes were found to trigger VF in 100% of cases [52].

Triggered activity can result from afterdepolarizations, i.e., voltage depolarizations (oscillations) that occur during the repolarization phase of the action potential (early afterdepolarizations, EAD) or during the diastolic phase, after completion of repolarization (delayed afterdepolarizations, DAD) [53]. If these voltage oscillations are large enough, they can trigger a full action potential. EADs are induced by reactivation of inward L-type calcium current, which is favored by prolongation of action potential duration. Alternatively spontaneous calcium release from the sarcoplasmic reticulum (SR) can activate sodium/calcium exchanger (NCX) generating inward current while exchanging one Ca^2+^ out for 3 Na^+^ ions in. DADs are induced by NCX activity and are favored by increased diastolic Ca^2+^ concentration and increased NCX activity. EAD and DAD are quite similar, since both are favored by cardiomyocyte Ca^2+^ overload and NCX overactivity and can be triggered by spontaneous SR Ca^2+^ release [54,55]. Beta adrenergic stimulation may favor both EADs (through phosphorylation and activation of L-type calcium channels) and DADs (through phosphorylation and inducing SR Ca^2+^ leak). Downregulation of sarcoplasmic reticulum Ca^2+^-ATPase (SERCA) has similar pro-arrhythmic effect [55].

Abnormal automaticity of working cardiac myocytes can be induced by increased extracellular K+ concentration (resulting e.g., from ischemia), reduced expression/activity of potassium channels or increased expression/activity of ion channels responsible for inward depolarizing current (e.g., HCN channels).

Arrhythmic substrates involve factors that (1) slow the intracellular conduction, (2) increase the spatial dispersion of repolarization. Slow intercellular conduction favors reentry, since it provides time for recovery of excitability [56]. Intercellular conduction depends on two factors: amplitude and upstroke velocity of cardiomyocyte action potential (a function of activation of sodium channels, SCN5A) and expression and function of gap junctions consisting of connexins that provide electrical connection between adjacent cells. On the other hand increased spatial dispersion of repolarization favors development of unidirectional block, a prerequisite of reentry.

## 6. Ivabradine and Ventricular Arrhythmias

### 6.1. Preclinical Evidence

Preclinical studies of ivabradine are summarized in Table 1. Briefly, ivabradine reduced ventricular arrhythmias (VA, both ventricular tachycardia [VT]/ventricular fibrillation [VF] and premature ventricular complexes [PVC]), arrhythmic deaths and total mortality in the rat model of non-reperfused acute MI, both when given as prophylaxis [25] and within the first minutes of ischemia [57] and was as effective as metoprolol in the latter study. Furthermore ivabradine increased VF threshold during acute myocardial ischemia in the pig [58,59], reduced VA during ischemia and reperfusion in the isolated rat heart [60]. Finally ivabradine was as effective as metoprolol in reducing VA induction rate and mortality after four weeks of administration [61] in post-MI rats and moreover, we showed that ivabradine reduced VA induced by dopamine infusion in a rat model of post-MI CHF [62]. Furthermore it almost completely abolished PVC after 3 months of administration [63] in the same model. In transgenic mice with CHF ivabradine reduced both the incidence of VT and PVC as well as isoproterenol-induced VF and improved survival [30].

Ivabradine reduced inducible VA in a Langendorff-perfused rabbit heart in the model of short-QT syndrome induced by a potassium channel opener, pinacidil [66] as well as digitalis-induced VA in the same rabbit model [67], while it was proarrhythmic in a model of long QT syndrome [68].

Last but not least, ivabradine was ineffective in a mouse model of catecholaminergic polymorphic ventricular tachycardia (CPVT) [69].

### 6.2. Potential Mechanisms of Ivabradine Effects

In the rat model of acute non-reperfused MI we showed that ivabradine prevented dispersion of electrical and biochemical properties between the infarct borderzone and the remote ventricular myocardium, essentially preventing early (within 45 min) ischemia-related shortening of AP and late (24 h) prolongation of AP related to downregulation of potassium channels (ERG and KVLQTI) in the infarct borderzone [25]. Despite the fact that ivabradine prevented striking differences in expression of various potassium channels (Kv4.3, KChIP2, ERG and KVLQTI) between the infarct borderzone and remote myocardium, it resulted in overall reduced expression of potassium channels, which could be responsible for QT prolongation observed at 24 h. Despite the fact that ivabradine did not affect parameters of Ca^2+^ handling, such as SR Ca^2+^ content, activity of major Ca^2+^ transporting proteins (NCX, sarcoplasmic Ca^2+^-ATPase [SERCA]), it reduced RyR sensitivity to Ca^2+^, potentially preventing pro-arrhythmic diastolic SR Ca^2+^ leak [25,57]. Moreover, in another rabbit ischemia and reperfusion study ivabradine increased FKBP12/12.6 expression and reduction of diastolic SR Ca^2+^ leak [70]. Another fascinating effect of ivabradine is stimulation of glycolysis and inhibition of hexosamine biosynthetic pathway in a HR-independent manner [71]. This in turn reduces O-linked N-acetylglucosamination of Ca^2+^-Calmodulin dependent protein kinase II and consequently reduces RyR sensitivity to Ca^2+^ [72]. However, ivabradine was ineffective in the model of CPVT: it did not prevent abnormal Ca^2+^ transients or arrhythmias resulting from abnormally increased RyR sensitivity in a transgenic mouse model of CPVT in vivo or in human induced pluripotent cells, exposed to isoproterenol [69].

Another potentially antiarrhythmic effect of ivabradine identified in our study was prevention of HCN4 overexpression in the ventricular cardiomyocytes as early as 24 h after MI induction [25], corroborating previous findings that ivabradine prevented HCN4 overexpression after 3 months of treatment in the post-MI rat [27]. However, due to the fact that in the setting of acute ischemia and reperfusion arrhythmias tend to occur early, before changes in ion channel expression could take place, [25,57], HCN4 overexpression is unlikely to play a major role in arrhythmogenesis in this setting.

In the pig model of acute ischemia [64] and ischemia and reperfusion [58,59] ivabradine increased regional blood flow, reduced the extent of myocardial ischemia, preserved myocardial energy status and prevented AP shortening [58], which was associated with protection against induced VF [58,59] and prolongation of time to onset of spontaneous VF [64]. In the latter study ivabradine was much more effective than propranolol in preventing spontaneous VF. Pacing to maintain unchanged HR essentially abolished protective effects of ivabradine, suggesting that HR reduction is the principal mechanism of its antiarrhythmic action [64]. Other experimental studies support this conclusion: we showed that ivabradine and metoprolol provided similar protection against VA induced by acute non-reperfused MI when given at doses that ensured identical HR reduction [57]. Similarly, ivabradine increased time to onset of ischemia-related conduction slowing and loss of electrical excitability and protected against development of VF during reperfusion in a model of 8-minute ischemia and reperfusion in an isolated rat heart [60]. Pacing throughout ischemia-reperfusion that completely prevented ivabradine-induced HR reduction completely abolished the protective effects of ivabradine, whereas pacing at reperfusion only partially attenuated this effect (by 40%). When ivabradine was given only at the beginning of reperfusion, it had no effect on the development of VF. The above mentioned studies strongly suggest that the principal mechanism of anti-arrhythmic effects of ivabradine in the setting of acute and chronic ischemia with/without reperfusion is prevention of ischemia itself related to HR reduction rather than specific effects on cardiac electrophysiology.

CHF is another highly proarrhythmic condition. A series of elegant studies were performed in a model of transgenic mice (dnNRSF-Tg), where CHF was induced by expression of a cardiac-specific dominant-negative form of neuron-restrictive silencer factor, an important regulator of the fetal cardiac gene program, characterized by dilated cardiomyopathy and sudden arrhythmic deaths [30,31]. In this model ivabradine prevented spontaneous PVC and VT/VF, but did not prevent pacing induced VT, indicating that it reduced the trigger rather than the substrate of ventricular arrhythmias. HCN2 and HCN4 were overexpressed and *I*_f_ amplitude was increased in this model. Moreover, ivabradine did not affect HR or the sympathovagal balance (as indicated by no change of HRV), again supporting the notion that reduction of an arrhythmic trigger is the principal driver of beneficial antiarrhythmic ivabradine effects and that this action is unrelated to HR reduction. Resting plasma membrane potential was slightly depolarized, which was unaffected by ivabradine, and presumably related to reduced *I*_K1_ density. Ivabradine reduced spontaneous action potentials induced by isoproterenol. However, these results need to be interpreted with caution: a model of CHF in the transgenic mice induced by reactivation of the fetal cardiac gene program is characterized by more than 60% arrhythmic mortality by week 28 of life. This model may overestimate role of HCN4 and arrhythmias in the CHF pathogenesis and mortality, respectively; its relevance for the human CHF is unknown [31]. In another studies transgenic ventricular myocytes with HCN2 overexpression exhibited ectopic activity induced by isoproterenol [30,73] and hypokalemia [74] that was reduced by ivabradine. However, HCN2 expression was increased 100-fold, while human failing ventricular myocytes demonstrate twofold to threefold increase of HCN expression. This raises questions of physiological relevance of this model.

Transgenic mice that expressed human HCN4 (hHCN4) under control of the murine cardiac troponin I gene promoter, providing moderate increase of *I*_f_ in cardiac myocytes to levels observed in human heart failure were used in another study [65]. It produced a phenotype of dilated cardiomyopathy, which was prevented by ivabradine. HCN4 overexpression resulted in diastolic cardiomyocyte depolarization (wild type: P_50_ = −61 mV vs. HCN4^tg^ P_50_ = −54 mV). The amplitude of AP overshoot was diminished in transgenic cardiomyocytes, suggesting an inactivation of voltage-gated Na^+^-current at more depolarized resting membrane potentials. AP duration at 20 (APD_20_) and 50% (APD_50_) repolarization were shorter in transgenic cells, while APD_90_ did not differ between wild type and HCN4^tg^ cardiomyocytes. This was accompanied by intracellular Ca^2+^ accumulation: increased diastolic Ca^2+^ concentration, SR Ca^2+^ content, and Ca^2+^ transients. All these abnormalities of cardiomyocyte Ca^2+^ handling could be blocked by a reverse mode NCX inhibitor, indicating that intracellular Na^+^ accumulation due to its inflow through HCN4 favors reverse mode NCX activity and cellular Ca^2+^ accumulation. HCN4^tg^ cardiomyocytes were prone to afterdepolarizations resulting in trains of premature APs. However, the increased automaticity most probably originated from *I*_f_ augmentation rather than from spontaneous Ca^2+^ cycling-mediated membrane depolarization, as the NCX equilibrium was shifted toward reverse mode leading to an outward, hyperpolarizing current. This pathological automaticity was blocked by ivabradine. Unexpectedly, during telemetric recordings HCN4^tg^ animals exhibited no sustained arrhythmias, only higher numbers of PVCs and few nonsustained VTs indicated increased arrhythmogenicity compared to wild type animals. These arrhythmias were blocked by ivabradine. This study suggests that upregulation of *I*_f_ in CHF may be proarrhythmic in the mechanism of increased automaticity, but this trigger alone is not sufficient to induce life threatening, clinical malignant arrhythmias [65].

As mentioned above, experimental studies suggest that HCN4 overexpression could play a role not only in arrhythmogenesis, but also in progression of CHF itself [65]. However, we have recently demonstrated that ivabradine prevented HCN4 overexpression in the rat model of chronic MI-induced HF, but unexpectedly a two-week dopamine infusion that had detrimental effects on both cardiac hemodynamic and remodeling parameters as well as induced ventricular arrhythmias, was also associated with reduced HCN4 overexpression [62]. Furthermore the degree of HCN4 overexpression in a large group of explanted human failing hearts did not correlate with severity of ventricular arrhythmias or hear failure [62].

Ivabradine was also tested in a setting of drug-induced electrophysiological changes in the rabbit heart (Table 1) [66,67,68]. A consistent pattern can be found in these studies: ivabradine prevented proarrhythmic effects of drugs that shorten QT interval (i.e., a potassium channel opener, pinacidil [66] and ouabain [67]), while exacerbated arrhythmias when combined with drugs that prolong QT interval (sotalol and veratridine [68]). What is interesting, in all these studies ivabradine only mildly increased APD or QT interval, while markedly increased effective refractory period, in particular post-repolarization refractory period, which could suggest that both *I*_Kr_ inhibition and HCN blockade were responsible for this effect. Since ivabradine was given at a relatively high dose (5 µM), again clinical relevance of this finding is uncertain.

### 6.3. Clinical Evidence

We must emphasize that there is still inadequate clinical evidence regarding ivabradine effects on cardiac ventricular arrhythmias.

For more detailed discussion of the clinical evidence the reader is referred to our previous review [75]. Briefly, three large randomized, placebo controlled clinical trials (BEAUTIFUL, SIGNIFY and SHIFT) and their Holter substudies assessed ivabradine in two indications: chronic CAD and CHF in a total of more than 36,000 patients [76,77,78,79,80]. The incidence of VA, namely VT or PVC did not differ between ivabradine and placebo groups. However, patients with chronic CAD have low risk of VA, while patients with CHF and high arrhythmic risk were excluded from those trials. This data is summarized in Table 2. Smaller trials enrolling patients with acute myocardial MI [81,82,83] also did not find any effect of ivabradine on VA, while in smaller studies enrolling patients with CHF [84,85,86] ivabradine reduced PVC, though no VT was recorded.

In two case reports CPVT ivabradine successfully suppressed VPC and nsVT [87,88].

There are no reported effects of a plethora of mutations of HCN channels on ventricular arrhythmias in humans: supraventricular bradycardia/tachyarrhythmias predominate [89].

There are anecdotal reports of proarrhythmic effects of ivabradine, in the setting of hypokalemia [83] and co-administration with QT-prolonging drugs [50,51].

Thus no robust clinical data is available on effect of ivabradine on ventricular arrhythmias in either CAD or CHF.

## 7. Possible Future Applications of Ivabradine

Ivabradine, the first in class and so far the only approved HCN channel blocker, can potentially be used in other arrhythmias, too. Two areas close to clinical application are rate control in atrial fibrillation (in combination with a beta blocker) [90] and treatment of pediatric junctional tachycardia [91,92].

## 8. Summary

Ivabradine is a HR reducing agent, currently approved for the treatment of CAD and CHF. Experimental studies suggest that ivabradine may have some antiarrhythmic effects: (1) in the setting of ischemia that may be secondary to HR reduction and consequent energy preservation and prevention of ischemia-induced electrophysiological effects and (2) in the setting of CHF that may be secondary to prevention of HCN channel overexpression and/or blockade of HCN channels and consequent reduction of proarrhythmic pathological automaticity in ventricular cardiomyocytes. However, ivabradine is probably also a blocker of I*_Kr_* and may prolong QT and induce VT under conditions of long QT (e.g., drug-induced). No robust clinical data is available on effect of ivabradine on VA in either CAD or CHF. Clearly further molecular studies are required to further elucidate such issues as ivabradine specificity, dose-related effects, electrophysiological effects in normal and diseased heart as well as clinical outcomes.

## Figures and Tables

**Figure 1 jcm-10-04732-f001:**
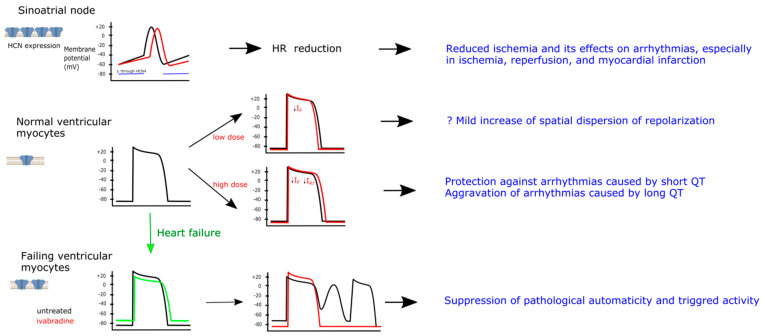
Electrophysiological effects of ivabradine in the heart.

**Table 1 jcm-10-04732-t001:** Effects of ivabradine on ventricular arrhythmias in experimental studies.

Species	Model	Study Design, Route and Method of IVA Administration	Effects on VA	Additional Effects	Reference
**Ischemia and reperfusion, myocardial infarction**					
Rat	non-reperfused MI, LAD ligation, first 24 h	IVA vs. saline, 24 h before MI, IVA in drinking water, daily dose 10 mg/kg	↓mortalilty ↓VT/VF frequency ↓VT/VF duration ↓VPC	↓RyR sensitivity ↓HCN4 expression in LV ↓dispersion of APD	[25]
Rat	non-reperfused MI, LAD ligation, first 6 h	IVA vs. metoprolol vs. saline, given immediately after MI induction, IVA 5 mg/kg oral gavage	↓mortalilty ↓VT/VF frequency ↓VT/VF duration ↓VPC as effective as metoprolol	↓QTc duration ↓RyR sensitivity -velocity of conduction -repolarization	[57]
Rat	non-reperfused MI, after 1 month	IVA vs. metoprolol vs. IVA + metoprolol vs. saline, starter immediately after MI and given for 4 weeks after MI, 10 mg/kg oral gavage	↓VT/VF inducibility and VT/VF fatality, IVA+Metoprolol better than IVA or metoprolol alone, which were better than saline	↓infarct size and ↑connexin 43 expression in all treatment groups	[61]
Rat	non-reperfused MI, months 3–5	IVA vs. saline, started 2 months after MI and given for 3 months, 10 mg/kg in drinking water	↓PVC by 89%	↑HR variability by 22% -PR, QRS, QT duration	[63]
Pig	6 episodes of I/R (LAD occlusion): 1 min ischemia + 15 min reperfusion	IVA vs. saline, given 5 min after second occlusion, IVA 0.25 mg/kg intravenous bolus	↑VF threshold by 2.9-fold	prevention of ischemia-induced APD shortening ↓hypoxia ↑regional blood flow	[58,59]
Pig	ischemia (LAD occlusion)	IVA vs. propranolol vs. saline, given before ischemia, IVA 0.25 mg/kg intravenous bolus	↓time to VF onset (2325s for IVA vs. 682s for propranolol vs. 401s for saline), abolished by pacing	Preserved cardiac energy status	[64]
Rat	isolated hearts, I/R episodes: 8 min of ischemia and 10 min of reperfusion	IVA vs. saline, 5 min before ischemia 1 µM IVA and 1ml bolus of 100 µM IVA before the first minute of reperfusion	↓VF incidence (IVA: 20%, saline: 90%) effect abolished by pacing throughout I/R, but not during reperfusion alone No effects when IVA given at reperfusion	↑time to conduction slowing ↑time to loss of electrical excitability	[60]
**Chronic heart failure**					
Rat	post-MI CHF	1 month after MI: IVA 7 µg/min/kg or Dopamine 10 µg/min/kg or both by implantable osmotic pump for 14 days	↓VT incidence and ↓PVC by IVA vs. Dopamine		[62]
Mouse	transgenic mice (dnNRSF-Tg) with dilated cardiomyopathy	IVA vs. saline given for 6 months, daily dose 7 mg/kg in drinking water	↓mortality ↓VT episodes (19/h vs. 92 h for saline) ↓VPC effects independent of HR reduction	↓HCN2 and HCN4 expression in ventricular myocytes IVA prevented isoproterenol-induced spontaneous action potentials in failing ventricular myocytes	[30]
Mouse	transgenic mice (hHCN4 overexpression) with dilated cardiomyopathy	IVA vs. saline given for 2 months, daily dose 1.5 mg/kg by an osmotic pump	↓VPC ↓non-sustained VT	↓automaticity	[65]
**Other models**					
Rabbit	pinacidil-induced short QT syndrome	Langendorff-perfused hearts, IVA 5 µM	↓VF inducibility	IVA prevented QT, APD and ERP shortening by pinacidil	[66]
Rabbit	ouabain-induced arrhythmias	Langendorff-perfused hearts, IVA 5 µM	↓VF inducibility	IVA did not affect QT or APD, but prevented ERP shortening by ouabain	[67]
Rabbit	sotalol and veratradine-induced long QT syndrome	Langendorff-perfused hearts, IVA 5 µM	↑incidence of polymorphic VT induced by sotalol/vertatridine and hypokalemia	IVA did not increase QT, APD or spatial dispersion of repolarization	[68]
Mouse	CPVT transgenic mice	IVA 6 mg/kg	-VA at rest or during a treadmill exercise test	no effect on DAD in the beating induced pluripotent stem cells derived cardiomyocytes from CPVT patients	[69]

APD, action potential duration; CHF, chronic heart failure; CPVT, catecholaminergic polymorphic ventricular tachycardia; I/R, ischemia and reperfusion; IVA, ivabradine; LAD, left anterior descending artery; MI, myocardial infarction; PVC, premature ventricular complexes; RyR, ryanodine receptor; VA, ventricular arrhythmias; VF, ventricular fibrillation; VT, ventricular tachycardia. ↑ increase; ↓ decrease; - no change.

**Table 2 jcm-10-04732-t002:** Effects of ivabradine on ventricular arrhythmias in large clinical trials.

Study Acronim, Year	Study Population	No. of Study Subjects	Follow-Up	Effect on VA
**Coronary artery disease**				
BEAUTIFUL 2008 [76]	CAD + LV systolic dysfunction, sinus rhythm ≥ 70 bpm	10,917	1.6 years	
BEAUTIFUL 2008 Holter Substudy [77]		840	2 × 24 hours at 1 and 6 months	VT: ΔIVA: 24–26%, PLA: 26–24% (*P* = NS)
SIGNIFY 2014 [78]	CAD without CHF, sinus rhythm ≥ 70 bpm	19,107	2.3 years	Severe VA: ΔIVA: 0.8%, PLA: 0.7% (*P* = NS) ΔQT prolongation: ΔIVA: 1.8%, PLA: 0.7% (*P* < 0.001)
**Chronic heart failure**				
SHIFT 2010 [79]	CHF, LVEF ≤ 35%, sinus rhythm ≥ 70 bpm	6558	1.9 years	
SHIFT Holter Substudy 2010 [80]		602	24 h at 8 months	nsVT: ΔIVA: 28%, PLA: 33% (*P* = NS) ΔPVC: ΔIVA: 78/h PLA: 69/h (*P* = NS)

CAD, coronary artery disease; IVA, ivabradine; LV, left ventricle; LVEF, left ventricular ejection fraction; nsVT, nonsustained ventricular tachycardia; PLA, placebo; PVC, premature ventricular contractions; RCT, randomized clinical trial; VA, ventricular arrhythmias; VT, ventricular tachycardia.

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
