# Peer review of "Effect of Ivabradine on Cardiac Ventricular Arrhythmias: Friend or Foe?"

_jcm, 2021, doi:10.3390/jcm10204732_

Round 1

Reviewer 1 Report

The report of Oknińska et al, even if refers a remarkable amount of physiological data of Ivabradine, only reports the results obtained in animal studies, without any reference to the clinical events happening in humans.

Author Response

Thank you for this comment. Indeed this is a more basic science oriented review. We have referred readers to our recent review focusing on effects of ivabradine on cardiac arrhythmias that was more clinically oriented. Moreover we have rewritten the section devoted to ivabradine effects in the clinical setting to make it more informative. 

Reviewer 2 Report

In this review, Dr. Oknińska and colleagues discuss the basic and clinical properties of ivabradine and its applications in the cardiovascular disease management. Overall, this is a very nice and informative review. Nonetheless, I have some suggestions to be incorporated:

  • Page 2: The authors need to explain the contribution of funny current in both membrane and calcium clock of SAN. Please mention specifically these two clocks in the text.
  • "Ivabradine, a selective heart rate reducing agent, an inhibitor of HCN channels, may be one of such options." I am not sure if ivabradine can be classified as "selective" since it also affects other channels, as the authors have exemplified in the text, supported by other publications (PMID: 31169990 and PMID: 25911606, for example). Please comment on this.
  • The text underneath the figure in Page 2 is unclear. Is that a figure caption or another paragraph? Because it is too lengthy for a figure caption and some of the contents are redundant and shared similarities with the following texts. Also, there is no "Figure 1" label in it and there is no figure title so I am not sure what it is actually. 
  • "In such setting IF current may be responsible for abnormal automaticity in the ventricular cardiomyocytes (in particular when repolarization reserve is reduced), resulting in ventricular premature complexes that may trigger ventricular arrhythmias." I am wondering how funny current could affect the generation of PVCs? Is it pure automaticity-associated PVCs or also afterdepolarization-induced PVCs? Perhaps the authors could elaborate the mechanisms.
  • It would be valuable to also add a table summarizing the clinical studies on ivabradine as the authors did for the preclinical studies.
  • "Pathophysiology of vast majority of ventricular arrhythmias involves (1) a trigger and (2) a substrate that propagates the trigger." The authors also need to explain about the "driver" of arrhythmia. We know that in arrhythmogenesis, the 3 major components are substrate, trigger and driver. Please add some information about arrhythmia driver in light of ivabradine.
  • In Page 7, when explaining the pathophysiology, I think the authors would benefit from this review (PMID: 32188566). Consider adding some information from that review to complement the currently available information.
  • The summary is too long. I think the authors can remove some specific details from the summary to shorten this part and also to avoid unnecessary redundancy. Just focus on the key messages of the review.
  • I think there is a substantial overlap between the "Ivabradine and ventricular arrhythmias" and "Effect of ivabradine" parts. Consider merging them into one section to improve clarity.
  • It would be valuable to have a section about the potential future applications of ivabradine or other funny current blocking agents in complex cardiovascular diseases. Consider adding this new section before the summary.
  • Please perform a recheck for potential typos and grammatical errors. I will provide some examples below:
    • Abstract: "Cardiac life threatening ventricular arrhythmias, such as ventricular..." There is no need to mention "cardiac" because it is clear that VA occurs in the heart.
    • "Another option, implantable cardioverters defibrillators, are (is) costly and (an) invasive devices that requires regular monitoring"
    • "IF" should be with small "f" - "If". Please revise.
    • "presumably due to IKr blockade, although at high con-centrations (10 μM, while normal ivabradine plasma concentrations in humans following oral dosing 5 mg - 20 mg are 0.03 - 0.13 μM" the closing bracket is missing here.
    • "we indeed demon-strated that hEGR abundance is reduced in post-MI rat hearts as early as 24 hours after MI induction" maybe this should be "hERG"?
    • "In the rat model of acute non-reperfused MI, in the rat we showed that..."

Round 2

Reviewer 2 Report

First, thank you for addressing my previous comments and suggestions.

I would encourage the authors to improve the writing style. Although the manuscript is readable, it can still be significantly improved and by doing so, the readability and clarity of the manuscript could be increased. An assistance from a professional scientific writer could be helpful.

Regarding the proposed table about the effects of ivabradine on ventricular arrhythmias in clinical studies, I do still think that it is important to include because the authors are currently submitting this manuscript to JCM, which is a clinical journal, thus clinical data has to be the highlight of the manuscript. 

I have also (re)checked the extensive caption of Figure 1 and I do think that some of those sentences can be moved to the main text and then linked to Figure 1. For example, the authors can move the rest of the sentences starting from "Antiarrhythmic effects of ivabradine in the setting of ischemia ..." to the main text but please do a thorough check whether the information has been delivered elsewhere to avoid redundancy. 

Author Response

Thank you for your valuable comments and suggestions.

The manuscript was read and corrected by a native speaker.

The table with large clinical trials of ivabradine discussing potential effects of ivabradine on arrhythmias was added.

The caption of Figure 1 was significantly shortened.

Thank you once again for your comments